# A Systematic Review of Amino Acid PET Imaging in Adult-Type High-Grade Glioma Surgery: A Neurosurgeon’s Perspective

**DOI:** 10.3390/cancers15010090

**Published:** 2022-12-23

**Authors:** Raffaele De Marco, Alessandro Pesaresi, Andrea Bianconi, Michela Zotta, Désirée Deandreis, Giovanni Morana, Pietro Zeppa, Antonio Melcarne, Diego Garbossa, Fabio Cofano

**Affiliations:** 1Neurosurgery Unit, Department of Neuroscience “Rita Levi Montalcini”, “Città della Salute e della Scienza” University Hospital, University of Turin, 10124 Turin, Italy; 2Nuclear Medicine Division, Department of Nuclear Medicine, “Città della Salute e della Scienza” University Hospital, University of Turin, 10124 Turin, Italy; 3Division of Neuroradiology, Department of Diagnostic Imaging and Radiotherapy, “Città della Salute e della Scienza” University Hospital, University of Turin, 10124 Turin, Italy

**Keywords:** glioma, glioblastoma, PET-integrated resection, amino acid PET imaging, supramaximal resection

## Abstract

**Simple Summary:**

Several advantages of molecular imaging, namely, positron emission tomography (PET), have been already described in different settings of glioma management. Particularly, the use of amino acidic radiotracers for PET imaging has gained favor for their added value in diagnosis, grading, guidance, and response to treatment and in order to rule out recurrences. Despite the meaning of the biologically active volume of the tumor, a PET-integrated resection of adult-type diffuse high-grade gliomas is not routinely performed, but it can represent a further perspective for neurosurgeons. A systematic review of the literature has been performed to investigate this topic with a precise focus on the neurosurgeon’s point of view presented, examining the reasons of its limited use in surgical practice and possible applications.

**Abstract:**

Amino acid PET imaging has been used for a few years in the clinical and surgical management of gliomas with satisfactory results in diagnosis and grading for surgical and radiotherapy planning and to differentiate recurrences. Biological tumor volume (BTV) provides more meaningful information than standard MR imaging alone and often exceeds the boundary of the contrast-enhanced nodule seen in MRI. Since a gross total resection reflects the resection of the contrast-enhanced nodule and the majority of recurrences are at a tumor’s margins, an integration of PET imaging during resection could increase PFS and OS. A systematic review of the literature searching for “PET” [All fields] AND “glioma” [All fields] AND “resection” [All fields] was performed in order to investigate the diffusion of integration of PET imaging in surgical practice. Integration in a neuronavigation system and intraoperative use of PET imaging in the primary diagnosis of adult high-grade gliomas were among the criteria for article selection. Only one study has satisfied the inclusion criteria, and a few more (13) have declared to use multimodal imaging techniques with the integration of PET imaging to intentionally perform a biopsy of the PET uptake area. Despite few pieces of evidence, targeting a biologically active area in addition to other tools, which can help intraoperatively the neurosurgeon to increase the amount of resected tumor, has the potential to provide incremental and complementary information in the management of brain gliomas. Since supramaximal resection based on the extent of MRI FLAIR hyperintensity resulted in an advantage in terms of PFS and OS, PET-based biological tumor volume, avoiding new neurological deficits, deserves further investigation.

## 1. Introduction

Among adult-type high-grade gliomas, glioblastoma (GBM) IDH wildtype is the most common malignant brain tumor in neuro-oncology practice [1,2]. Maximal safe resection is the first-line treatment, followed by concomitant chemoradiotherapy (Stupp protocol [3]). Different studies have demonstrated the importance of the extent of resection (EOR) in terms of progression-free survival (PFS) and overall survival (OS) [4]. Generally, the extent of resection in newly diagnosed GBM aims to reach the contrast-enhanced borders of the lesion, which is defined in contrast-enhanced magnetic resonance imaging (ceMRI) [5]. Although there is no unique definition of gross total resection (GTR) [6], most studies report a complete removal of tumor at postoperative MRI stating contrast-enhanced tumor as a benchmark for GTR. This goal is affordable, taking advantage of different tools, such as fluorophores (i.e., sodium fluorescein and 5-aminolevulinic acid (5-ALA), which are the most used and validated) [7,8] or intraoperative imaging (i.e., neuronavigation system, intraoperative MRI (iMRI), intraoperative ultrasound (iOUS), and contrast-enhanced ultrasound (CE-US)) [9,10,11,12,13,14]. More recently, supratotal or supramaximal resection (SupTR) has proven to increase PFS and OS [15,16,17,18]. It is defined as complete removal of signal abnormalities beyond the contrast-enhanced borders of the tumor, and its rationale is in the assumption that glioma cells are infiltrative a priori [19,20]. Indeed, the majority of relapses are at a tumor’s margins [21,22]. Further signal abnormalities beyond the contrast-enhancing nodule include the T2-weighted hyperintensity seen at fluid-attenuated inversion recovery (FLAIR) MRI sequence [23] or the biologically active areas marked in positron emission tomography (PET). Few comparisons between MRI and different PET-based imaging techniques have revealed the presence of an active tumor beyond contrast-enhanced borders, and sometimes active regions could be even placed outside the T2/FLAIR hyperintensity [24]. Amino acid PET (AA-PET) imaging has emerged as a reliable imaging technique, in terms of diagnosis, histopathological correlation, surgical planning, and prognosis [25]. With increasing specificity and sensitivity compared with 18F-fluorodeoxyglucose (18F-FDG) and other conventional imaging, radiolabeled amino acid (i.e., [methyl-11C]-L-methionine (Met), O-(2-[18F]fluoroethyl)-L-tyrosine (FET), 18F-fluoro-L-dihydroxy-phenylalanine (FDOPA)) PET can depict the metabolic activity of gliomas, highlighting areas with a different radiotracer uptake—and theoretically with a different malignant potential—regardless of the blood–brain barrier impairment, adding further information on tumor extension [26,27].

The purpose of the current review is to investigate the role of amino acid PET specifically in improving the extent of the resection of high-grade gliomas and accordingly increasing PFS and OS.

## 2. Materials and Methods

A literature review was conducted in the PubMed database investigating the use of PET imaging as a guidance in glioma surgery using the PubMed database and PRISMA (Preferred Reporting Items for Systematic Reviews and Meta-analyses) recommendations. In order to include as many relevant articles as possible, there were no restrictions on the date of publication. The search terms used were “PET” AND “glioma” AND “resection”, including all fields. The literature was systematically reviewed by two independent reviewers (RDM and AP). All disagreements were resolved by further discussion with the senior author (FC).

The selection process was characterized by the following inclusion criteria: (1) availability of the manuscript in English or an English translation, (2) primary clinical studies investigating the use of PET imaging to guide intraoperatively high-grade glioma resection, and (3) a population >18 years old. All articles reviewed were also subject to the following exclusion criteria: (1) case reports, (2) reviews and preclinical studies, (3) biopsies, (4) preoperative characteristics resembling low-grade glioma (LGG) and/or histological confirmation of LGG, (5) studies involving pediatric patients, and (6) nonuse of PET imaging data in the neuronavigation system as guidance for surgical resection.

The systematic review followed the recommendations of the Preferred Reporting Items for Systematic Reviews and Meta-Analyses (PRISMA). The protocol has not been registered.

## 3. Results

The PubMed search yielded 276 results after confirming the absence of duplicates.

Out of the 276 unique papers, full-text analysis was performed for 143 articles. In most cases, PET imaging was used for assessing the extent of resection. In 15 articles, PET imaging was integrated in surgical planning, but only one article met both inclusion and exclusion criteria [28] (Figure 1: PRISMA Flowchart).

Indeed, a study conducted by Inoue et al. [28] considered only adults affected by newly diagnosed high-grade glioma (WHO 2021 grade 4) that underwent surgical resection with the integration and fusion of Met-PET imaging during surgery. They demonstrated the presence of glioma stem cells beyond the border of the MRI enhancing nodule. Furthermore, an important part of the biologically active tumor, highlighted with the PET imaging technique, was located outside the borders defined by gadolinium. Specifically, an analysis of a different tumor-to-contralateral normal brain tissue ratio (TNR), from 1.2 to >2.0, found that the threshold at 1.4 was always beyond the borders of the MRI contrast-enhanced nodule, but, at the same time, in nine cases out of 10, was smaller than the area with FLAIR hyperintensity. A pathological investigation of tumor pieces obtained by these different areas confirmed the biological significance in terms of proliferative index (ki-67 of 23.6%, range of 5.8–68.3% vs. 39.3%, range of 14.9–68.0%, which was the one tissue obtained by the highest contrast-enhanced and highest SUV-PET-positive areas).

Conversely, the auxiliary use of PET imaging and its integration in the neuronavigation system as a tool to localize the bioptic area and to define the borders in primary diagnosed glioma was described in 15 articles (Table 1) [29,30,31,32,33,34,35,36,37,38,39,40,41,42,43].

Sample size, patient demographics (i.e., sex and age), extent of resection, overall survival, and progression-free survival were the information retrieved from the selected article (Table 2). An eligible article was excluded due to the inclusion of children in their study population [45].

## 4. Discussion

The results show a paucity of studies in which PET imaging was integrated in the surgical planning and used to guide resection beyond the contrast-enhanced borders of conventional MRI. Furthermore, a great discrepancy and heterogeneity among available articles was observed, both in the design and in the aim of the studies.

In the diagnostic setting, different reports have mainly correlated PET tracer uptake in glioma tissue with its biological characteristics, namely, cell density [47,48,49,50]. This correlation has been demonstrated to be stronger than that seen intraoperatively with 5-ALA [36,51].

The most used radiolabeled amino acid PET imaging shares the same mechanism to enter inside tumor cells: L-amino acid transporters (LAT) are broadly expressed in tumor tissue but not in normal brain tissue, increasing the tumor-to-background contrast compared with 2-deoxy-2-[18F] fluoro-D-glucose (FDG)-PET, which presents a high physiological brain uptake [52]. Specifically, LAT1 is the main transporter type for Met, FET, and FDOPA [53,54,55]. Once inside glioma cells, the metabolic pathway differs for each one of them, opening on different possibilities of clinical investigations and applications [56].

In addition to the specific molecule used as radiopharmaceutical, the radiolabeling process and the involved radioisotope are different and have an impact on image acquisition and availability. In fact, 11C amino acids show a short half-life (only 20 min) compared with 18F radiolabeled amino acids (110 min), needing an on-site cyclotron [56] and limiting their use to a few centers. The introduction of 18F radiolabeled amino acids in brain tumors—especially in glioma—busted an increasing interest in these radiotracers with some advantages compared with Met-PET imaging [56]. Indeed, a rapid research on PubMed for “FET-PET AND Glioma” returned nearly 200 articles in the period between 2015 and 2020. Although with less impact compared with FET-PET, 18F-DOPA PET has gained interest in adult and pediatric glioma research in the last decade [57,58,59,60,61,62,63].

All amino acids have shown to offer additional information to conventional imaging, but only few studies have compared them directly [64,65,66]. Beyond its role in diagnosis [44,67] and eventually in grading [52,63,68,69,70,71], AA-PET imaging can offer information on a tumor’s extension in a more specific and sensitive way than structural MRI with gadolinium and FLAIR sequence as well [24]. Furthermore, this additional information could result in paramount importance for surgical planning when a significant part of the tumor lacks the contrast enhancement [72,73,74].

Indeed, the calculation of biological tumor volume (BTV) could detect more meaningful areas (in terms of pathological diagnosis) when a biopsy is planned or when MRI information is not sufficient for safe resection, either because the MRI borders reach eloquent brain regions or because it might correspond to scar tissue in case of glioma recurrence [36,75]. Furthermore, the integration of AA-PET imaging for surgical planning or for biopsy is recommended by the current guidelines [26,27].

In detail, AA-PET imaging has been demonstrated to more accurately identify in-filtrating regions of tumor extending beyond the MRI contrast-enhancing lesion, delineating significantly larger tumor volumes, and to better define tumor boundaries within nonspecific regions of MRI T2/FLAIR signal abnormality (infiltrative disease vs. vasogenic edema) [76]. Additionally, it provides further insights regarding tumor heterogeneity, biological activity, or aggressiveness of the disease [77].

Most of the studies that have analyzed the integration of PET imaging in the operating room focused on the histopathological validation of PET findings. Conversely, a pioneering work by Pirotte et al. demonstrated an advantage of PET-guided resection for grade 3 and 4 gliomas [45]: a statistically significant difference of 14.9 months benefited those patients without postoperative PET tracer uptake.

Although a thoroughly demonstrated association between PET tracer uptake (Met-PET) and viable glial cells has been identified in the core of the tumor, the same correlation has not always preserved at the tumor border and especially at tumor-infiltrated areas. Other techniques have been proposed to increase the reliability of PET imaging in MRI nonenhanced areas (beyond the contrast-enhanced nodule) [78,79,80].

Inoue et al. [28] focused on the tumor-to-contralateral normal brain tissue ratio (TNR) to evaluate the metabolic activity of GBM. Using Met-PET, they demonstrated the presence of glioma stem-like cells at TNR 1.4 of the tracer uptake, beyond the contrast-enhanced borders of standard MRI. Specifically, dividing GBMs in three subtypes (A, B, and C types) on the basis of radiological features, they found a better progression-free survival and overall survival for those subtypes where the difference between PET-based borders’ evaluation (TNR of 1.4) and contrast-enhanced T1-weighted MRI imaging was low. Indeed, a GTR of contrast-enhanced borders as in the latter GBM subtype (namely, B subtype) was an indirect expression of meaningful PET radiotracer uptake area resection. Since most glioma recurrences occur at borders of the resected cavity, hypothesizing an origin of tumor recurrence from glioma stem-like cells, the surgical goal should seek these margins according to PET uptake when safe. However, the same limits should be used in the radiation treatment planning, as already proposed [81,82].

Nevertheless, the authors stressed the difficulty of safely performing a resection of this PET uptake area (characterized by TNR of 1.4), especially when the tumor arises near eloquent areas: the contemporary use of additional tools, such as intraoperative fluorophores and neuromonitoring, which enhances the information of a real-time image-guided navigation system, where both PET and MRI fusion images were used (i.e., echo-linked navigation and fence-post technique), allowed for obtaining a better outcome. From unpublished series, the median OS for patients where Met-PET imaging was integrated in the navigation system was of 21.6 months compared with 15.2 months where the same tools were used but without metabolic information. A similar improvement on survival was already reported for other series (which comprehended both children and adults and WHO grades III and IV), where PET imaging was integrated in surgical planning [45,83]. Even in the absence of an integrated PET imaging surgical procedure, a small postoperative BTV, more than GTR of the contrast-enhanced nodule, was associated with better rates of PFS and OS, without an increase in new neurological deficits [25].

A similar result was confirmed in a recent retrospective study where 30 patients with primary and recurrent WHO grade 3 and 4 gliomas were recruited [46] (this report was excluded due to the absence of a statement on the integration of PET imaging during resection). Patients with a primary diagnosed glioblastoma were 16 (overall 20). Performing an image analysis of preoperative MRI and FET-PET data, they found a BTV GTR in 20 patients, obtaining a statistically significant improvement in OS compared with the patients with PET residuals (19.3 vs. 13.7 months, *p* = 0.007), which remained significant in the group with a primary diagnosis of glioma (17.3 vs. 13.7 months, *p* = 0.048) and even for glioblastoma (15.1 vs. 13.6 months, *p* = 0.048).

Interestingly, Hirono et al. (this report was excluded for the same reason as above) analyzed retrospectively the survival benefit that should have been conferred by resecting additional tissue beyond contrast-enhanced borders that appeared to show Met-PET uptake. A resection beyond contrast-enhanced borders seen at standard MRI was considered a supratotal resection (SupTR). Among 30 patients with primary diagnosis of glioblastoma, they obtained 23 GTRs and 7 SupTRs. Twelve patients underwent awake craniotomy with cortical and subcortical direct electrical stimulation (DES), and remarkably, all SupTRs were registered in this group.

Although most studies have been conducted with Met-PET, other reports have shown that 18F radiolabeled amino acids (i.e., FET-PET and 18F-DOPA-PET) provided similar information in glioma patients [66,81]. Among Met, FET, and DOPA PET tracers, similar results in terms of accuracy have been reported regarding the ability to detect tumor boundary [66,84,85]. When compared with Met and FET, the main potential drawback of DOPA is its specific uptake in the striatum that may affect the ability to assess the involvement of the putamen and caudate nucleus [85,86]. On the other hand, DOPA uptake within the striatum gives the opportunity to further stratify tumoral uptake ratios through comparison with both the normal background levels and the striatum [87].

Another matter of research has been a possible link between AA-PET radiotracers and intraoperative fluorescent dyes. Indeed, a relationship between FET uptake and 5-ALA fluorescence has been demonstrated [88], and despite few described cases, FET uptake appeared to be more sensitive compared with intraoperative 5-ALA [34]. However, the absence of intraoperative fluorescence was related to low FET-PET uptake. The more fluorescing tissues was removed during surgery, the lower PET uptake was registered at postoperative imaging, with significant improvement in survival [25].

Despite the results on the usefulness of PET data and their integration in the surgical planning, the introduction of PET imaging in the routine surgical practice has still not been achieved.

Focusing on EOR, different tools, used as single or in combination, have been investigated and promoted in the last decades: fluorophores detected by specific microscope filters [7,89,90,91,92,93,94,95], iMRI [10,11], iUS [12], CE-US [13], fibertracking [96,97], neuronavigation systems [98], and intraoperative neuromonitoring [99].

Each one of these tools, with its own advantages and disadvantages, seeks to increase EOR while preserving neurological functions.

Leaving out the restrictions of a clinical facility to produce radiolabeled amino acids, a possible limitation in the diffusion of PET imaging as an integrated tool in surgical planning could be searched in the intrinsic disadvantage of the neuronavigation system. Despite the widespread diffusion of the latter tool, glioma resection cannot rely exclusively on images. Indeed, neuronavigation is burdened by few limits and errors, such as the brain shift after craniotomy or after dural opening or yet during tumor resection, decreasing the accuracy of space coordinates registered before skin incision [100].

A few authors who have investigated the role of PET imaging in glioma have performed bioptic samples before starting resection, immediately after dural opening in order to keep brain shift at a minimum [31,34,35]. In other cases, a Nelaton catheter was introduced under navigation guidance and anchored to the area that was planned for biopsy [36]. A few others have performed a biopsy before opening the dura [38,40] or before craniotomy [42], and there have been reports of nonlinear registration of intraoperative 3D ultrasound with preoperative FLAIR sequence using a specific algorithm, such as RaPTOR [39].

Live imaging, such as iMRI and/or iUS, can mitigate the problem, but it does not provide the same class of information supplied by PET imaging. Fluorescence-guided surgery can address immediately the neoplastic tissue inside the boundary or even beyond the standard MRI contrast-enhanced borders. Indeed, 5-ALA could aid the surgeon in reaching a supratotal resection, highlighting the glial cells outside the necrotic core of the tumor, while possibly increasing the risk of postoperative neurological deficit [15,101]. Despite this fact, 5-ALA carries a few limitations: in addition to economic issues, patient preparation, and possible side effects, the type of information that 5-ALA provides is different from those offered by radiolabeled amino acids [34,102].

Since the majority of studies have investigated the pathologic “meaning” of the AA-PET-positive tissue obtained with biopsy, only a few have analyzed the impact of performing a resection with the integration of AA-PET imaging: although PET-positive areas are often beyond the border of a contrast-enhanced nodule, no increase in neurological deficit has been reported.

Despite being mentioned in the “only biopsy” group, the multicenter clinical trial by Wakabayashi et al. [43] interestingly reported a modification of the extent of resection considered in the surgical planning of the use of 18-fluciclovine-PET in 47.2% of cases, increasing by 47.8% the resection of high-grade glioma.

Aiming for the resection of AA-PET-positive areas and BTV could be a good compromise between “FLAIRectomy” and classical GTR. However, as the functional result of the patient is the surgeon’s real aim, it is clear that an image-based resection has its well-known limitations [103,104,105].

## 5. Conclusions

Although an established role was achieved in different fields of glioma management, few data are currently available on PET-guided surgery in high-grade gliomas of the adult. However, with the emergence of the supramaximal resection concept and the lack of reliable radiologic methods to define the microscopic extent of the tumor, a preoperative PET role may assume greater relevance. In combination with other tools, such as MRI, neuronavigation, and fluorophores, great expectations could be reserved in supporting an EOR beyond macroscopically pathologic margins and in tumor border delineation, achieving a more safe and complete resection and an increasingly optimal oncologic–functional balance.

However, beyond the technical aspects pushing in this direction, there is a need for prospective studies evaluating the impact on overall survival and progression-free survival, confirming the validity of PET integration to multimodal neuroimaging in high-grade glioma surgery.

## Figures and Tables

**Figure 1 cancers-15-00090-f001:**
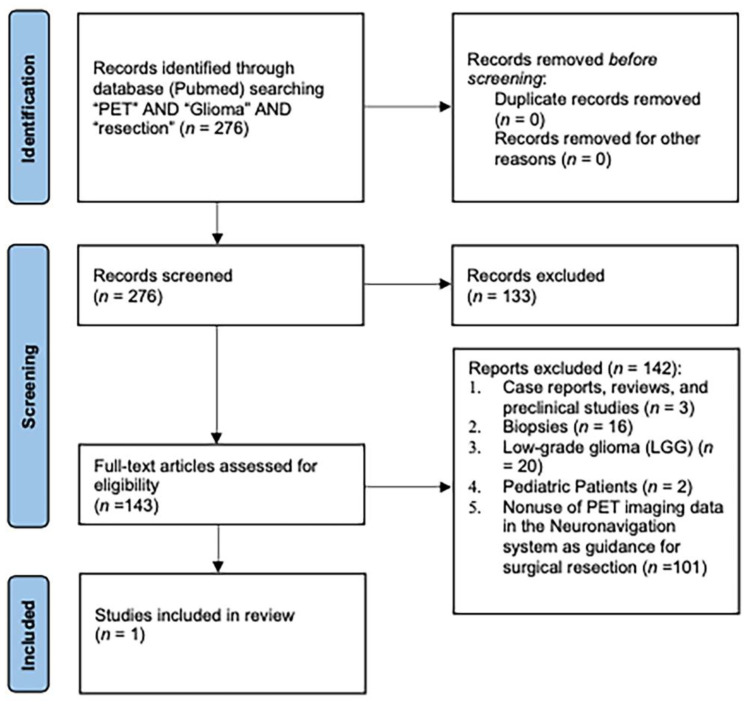
PRISMA 2020 flow diagram illustrating the study selection.

**Table 1 cancers-15-00090-t001:** List of references where PET imaging was integrated in the neuronavigation system and high-uptake areas were preferentially selected for biopsy.

Authors, Year	No. of Pts	No. of Recurrence	Study Type	PET Radiotracer	PET Use	Glioma Grade (WHO 2021)
Pirotte et al., 2004 [29]	32	0	Retrospective	Both 18F-FDG and Met	Comparison of distribution, extent, and relative contributions of 18F-FDG and Met	2–4
Pauleit et al., 2005 [44]	28	0	Prospective	FET	FET-PET combined with MRI imaging to improve distinction between cellular glioma tissue and unspecific peritumoral brain tissue	1–4
Stockhammer et al., 2007 [30]	25	4	Retrospective	FDG	Prediction of LOH 1p/19q in grade II gliomas	2, 3
Stockhammer et al., 2008 * [31]	22	9	Prospective	FET	Histological evaluation of surgically collected tissue in FET-PET-positive areas of non-contrast-enhancing lesions	2, 3
Weber et al., 2010 [32]	61 ^1^	0	Retrospective	FLT and FDG	Multiparametric evaluation and functional imaging comparison	2–4
Kunz et al., 2010 [33]	55	0	Prospective	FET	Correlation between dynamic PET parameters and histopathological characteristics, metabolic and molecular signatures	2–4
Floeth et al., 2011 * [34]	30 ^2^	4	Prospective	FET	Correlation between FET-PET uptake and intraoperative 5-ALA fluorescence	2, 3
Ewelt et al., 2011 * [35]	30 ^2^	NA	Prospective	FET	Relationship between FET-PET uptake, contrast-enhanced areas at MRI, and 5-ALA fluorescence intraoperatively	2, 3
Arita et al., 2012 * [36]	11	NA	Prospective	Met	Correlation between Met-PET uptake and intraoperative 5-ALA fluorescence	2–4
Pafundi et al., 2013 [37]	10	2	Prospective	18FDOPA	Histopathological differences between CE-MRI areas and 18FDOPA areas and correlation of pathological characteristics with PET uptake and use in RT planning	2–4
Beppu et al., 2015 * [38]	13	0	Prospective	FRP170	Comparison of FRP170-PET uptake areas with histological findings	4
Karlberg et al., 2019 * [39]	11	3	Prospective	F-FACBC	Diagnostic value of 18F-FACBC PET/MRI in distinguishing between low-grade and high-grade gliomas and its use in guiding surgical resection	2–4
Fernandez et al., 2019 * [40]	13	0	Prospective	FLT	Assessment of the added value of PET imaging integration to MRI in detecting tumoral tissue and correlation of PET uptake to tumor proliferation and grading	4
Ponisio et al., 2020 [41]	10	4	Prospective	18FDOPA	A better assessment of tumor volume and surgical margins and correlation of 18FDOPA-PET/MRI imaging with grade, histopathology, and molecular markers	2–4
Verburg et al., 2020 * [42]	20	0	Prospective	FET	Assessment of best imaging studies’ (both FET-PET imaging and different MRI sequences) combination to detect glioma infiltration in enhancing and nonenhancing glioma	2–4
Wakabayashi et al., 2021 * [43]	45	0	Multicenter, nonrandomized, open-label phase III clinical trial	18F-fluciclovine	Assessment of diagnostic accuracy of 18F-fluciclovine and useful in determining the extent of resection	2–4

* They provided information on methods to decrease the influence of brain shift in the area that was planned for biopsy. ^1^ Preoperative PET imaging was performed in only 20 patients. ^2^ Both papers presented the same series of patients; 18F-RP170: 1-(2-hydroxy-1-[hydroxymethyl]ethoxy)methy l-2-nitroimidazole (RP170); 18F-FACBC: anti-1-amino-3-[18F]fluorocyclobutane-1-carboxylic acid; FETNIM: fluoroerythronitroimidazole; 18F-fluciclovine: anti-1-amino-3-[18F]fluorocyclobutane carboxylic acid or anti-[18F]FACBC.

**Table 2 cancers-15-00090-t002:** List of references where AA-PET imaging integrated surgery for GBM was demonstrated to improve PFS and/or OS.

Authors, Year	No. of All Pts with HGG/No. of Primary HGG	Sex (All Pts)	Age (Mean; Range)	PET Radiotracers	PET Resection (y/n)	EOR	EOR Improvement (y/n)	PFS (Mean and Range)	OS (Mean and Range)	Limits
Pirotte et al., 2009 ^1^ [45]	66/31	23♀; 43♂	6–70	FDG (n. 9), Met (n. 22)	y	10/31 PET-STR 21/31 PET-GTR	y	NA NA	32.8 (NA)	3 children were included in the study; although the difference in overall survival was statistically significant, 32.6 or 32.4, compared with 17.6 months (χ2 = 20.231 (df, 1); *p* = 0.0001; median survival: R = 5.714; hazard ratio, 0.532), they did not report the OS for each group considered in the comparison
Inoue et al., 2021 [28]	10	3♀; 7♂	66.4 y (38–79)	Met	y	8/10 CE-GTR ^1^ 2/10 CE-STR	NA	17.5 mos (2.1–65)	26.4 mos (6–65)	
Ort et al., 2021; [46]	30/16	11♀; 19♂	59.9 y (53–63	FET	NA	12/20 PET-GTR 8/20 PET-STR	y	NA NA	15.1 13.6	Impossible to extrapolate the use of PET imaging

^1^ GTR was considered 100% resection of tumor volume, while a subtotal resection was considered between 95% and 100%; HGG, high-grade glioma; EOR, extent of resection; PFS, progression-free survival; OS, overall survival, Met, [methyl-11C]-L-methionine; CE-GTR, contrast-enhanced gross total resection; CE-STR, contrast-enhanced subtotal resection.

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
