# Peer review of "A Systematic Review of Amino Acid PET Imaging in Adult-Type High-Grade Glioma Surgery: A Neurosurgeon’s Perspective"

_cancers, 2022, doi:10.3390/cancers15010090_

Round 1

Reviewer 1 Report (Previous Reviewer 2)

The conclusion section says too much about the influence of this systematic review, as the authors admit that there are few pieces of evidence (Line 37) about the clinical use of the PET-imaging. This systemic review can provide integrated information on the current usage of the AA-PET from the viewpoint of the neurosurgeon but does not qualify as the source of the clinical practice.

Author Response

Thank you for taking the time to review our manuscript. We appreciate the opportunity to improve quality of our work through your insightful comments.

Since the main aim of the review was to focus on the utility and diffusion of PET imaging in surgical practice, we analyzed advantages, disadvantages and limits of this technique from a neurosurgeon’s point of view. Indeed, the surgical gesture and consequentially the EOR are until now what can change the OS of the patients.

However, considering the need of other therapies and the figures involved in the clinical management of patients affected with glioma, it is impossible to generalize the point of view of one with the clinical practice for such brain tumor.

Accordingly, we have modified our conclusion in the text.

Reviewer 2 Report (New Reviewer)

DeMarco, et al. performed a systematic review of amino acid PET imaging in adult high-grade gliomas surgery. A medical literature search was performed with regard to PET, glioma and resection with a hypothesis that amino acid pet imaging would be able to improve progression free survival and overall survival based on extent of resection. Only one study was identified as meeting the inclusion and exclusion criteria. However, a number of other references were listed and described in tables 2 and 3 reflecting a variety of amino acid-based PET tracers. The conclusion was that there was a paucity of studies in which pet imaging was integrated into surgical planning and used to guide resection beyond the contrast-enhanced borders of conventional MRI, and there was discrepancy and heterogeneity among available articles. The authors describe the pitfalls and advantages of using amino acid-based tracers for this indication, and noted that while some studies were showed possible efficacy, prospective studies were needed.

Comments:

Simple summary and abstract: Accurately reflects the methods, results and conclusions.

Introduction: Adequately describes the current methodology with regard to optimizing the neurosurgical resection of gliomas and the rationale for using amino acid pet.

Materials and methods: The inclusion criteria of English or English translation manuscript, primary clinical studies with use of PET imaging, adult population, and the exclusion criteria of case reports, reviews, preclinical studies, biopsy findings, low-grade gliomas, pediatric patients, and the use of non-pet imaging data in neuro-navigation systems are appropriate.

Results section: The description of the screening process was adequate. The single reference meeting inclusion criteria (Inoue et al) as detailed in the figure 1 flowchart were in should be described earlier in the results section. The authors listed other references that were not that did not meet criteria in tables 1 and two to demonstrate the type that demonstrated the types of studies the types and variability of studies that were previously performed.

The description of the findings in table 2 would be more readable if it was in a narrative and not footnote form.

Discussion: The discussion is as comprehensive as it can be, demonstrating the available technologies and their rationales. The inadequacy of the current state-of-the-art was discussed. The authors appropriately identified this type of technology as being promising with prospective studies needed. In the conclusions, the authors could perhaps indicate which what type of amino acid agents and methodologies would be most appropriate for future clinical trial and how such a trial should be performed. 

Author Response

Thank you for taking the time to review our manuscript, we appreciate your positive comments and the opportunity to improve the quality of our work.

Regarding results section, we have described the findings of Inoue et al. immediately in the section, giving more impact to this kind of results.

As you suggested we modified table 2 in order to make the table more readable removing the footnotes.

Regarding your suggestion in the conclusion, we think that the use of a specific amino acids for PET imaging should reflect the current guidelines “Joint EANM/EANO/RANO practice guidelines/SNMMI procedure standards for imaging of gliomas using PET with radiolabeled amino acids and [18F]FDG: version 1.0”

Reviewer 3 Report (New Reviewer)

The role of surgical removal in gliomas is important, and it is a well-known fact that the rate of resection determines the prognosis. The term “total resection” is appropriate when the border between tumor and normal can be distinguished, but this is not possible in gliomas.

Classically, the removal rate has been discussed based on what percentage of the contrasted tumor area could be removed on contrast-enhanced imaging studies. It is a well-accepted reality that glioma pathology has no clear borders and that tumor cells extend far beyond the tumor mass visible on MRI and CT contrast studies. Therefore, recently, " supra-total resection" has been attempted to remove more of the high signal of the second border, the T2-weighted MRI or FLAIR images, and the extension of life expectancy has been reported as a result of this approach. The tools that made this possible were monitoring and intraoperative fluorescence diagnosis. However, there is a limited extent within the brain that can be safely removed from a functional standpoint and ultimately cannot be removed by image guidance alone.

The authors are attempting to translate the previous literature regarding the use of PET imaging utilizing amino acid tracers to visualize tumor borders and use this hyper-accumulated area as an alternative to contrast-enhanced MRI/CT to plan resection and improve glioma prognosis by the rate of removal.

I agree with the authors' idea of using amino acid PET to visualize the border of the invasive tumor, which is considered to be in between removal of the contrasted area and removal up to T2/FLAIR high signal and fuse it with surgical navigation. As the authors state, this removal procedure is not easy, and even if it becomes visible, whether or not the area can be removed depends on its localization.

It is debatable which is better, the supra-total approach of removing as much of the T2/FLAIR high signal area as possible and the boundaries of this area are determined by monitoring, or the authors' approach of removing as much of the amino acid PET high accumulation border as possible.

Therefore, even though amino acid PET clearly visualizes the extent of tumor involvement, there seems to be a lack of discussion on the advantages and disadvantages of applying amino acid PET to surgical excision. There is no disagreement that amino acid PET indicates the detection of a tumor, but it would be better to use amino acid PET as an area to increase the intensity of following radiotherapy (e.g., IMRT), or to communicate the removal rate to patients in order to collect more cases as the amino acid PET hyper-accumulation range.

The manuscript lacks these perspectives and seems unlikely to be supported by nuclear medicine specialists who have been pushing for PET scans. Even though many of the statements made in the discussion section are traces of existing imaging studies, such as what amino acid PET is able to visualize, they do not support the authors' conclusion that it finds utility in determining its role in surgery, particularly in determining the extent of resection. It would be better to rewrite it broadly and make it useful in determining the extent of treatment, by citing, for example, manuscripts that have been useful in determining the extent of radiotherapy.

minor comments.

Only PRISMA flow is described, which is in accordance with the guide for systematic reviews, but the content and the way it is summarized is hardly a systematic review. It is more acceptable to simply refer to it as a "literature review".

The references picked up in Tables 1 and 2 should be detailed, as there is no indication which part corresponds from PRISMA.

Please give the references in the table a reference number.

It is unclear which one manuscript corresponded to the last remaining manuscript in PRISMA.

Author Response

 Reviewer #3:

Dear Reviewer, thank you for taking the time to review our manuscript and for your comments, we will try to answer to your concerns points by points.

The role of surgical removal in gliomas is important, and it is a well-known fact that the rate of resection determines the prognosis. The term “total resection” is appropriate when the border between tumor and normal can be distinguished, but this is not possible in gliomas.

Classically, the removal rate has been discussed based on what percentage of the contrasted tumor area could be removed on contrast-enhanced imaging studies. It is a well-accepted reality that glioma pathology has no clear borders and that tumor cells extend far beyond the tumor mass visible on MRI and CT contrast studies. Therefore, recently, " supra-total resection" has been attempted to remove more of the high signal of the second border, the T2-weighted MRI or FLAIR images, and the extension of life expectancy has been reported as a result of this approach. The tools that made this possible were monitoring and intraoperative fluorescence diagnosis. However, there is a limited extent within the brain that can be safely removed from a functional standpoint and ultimately cannot be removed by image guidance alone.

The authors are attempting to translate the previous literature regarding the use of PET imaging utilizing amino acid tracers to visualize tumor borders and use this hyper-accumulated area as an alternative to contrast-enhanced MRI/CT to plan resection and improve glioma prognosis by the rate of removal.

Since the only way to postoperatively assess the extent of resection is an imaging technique, although there is no consensus on the definition of gross total resection, each tool could be useful to a surgeon in order to improve the EOR balancing the functional outcome of the patients.

In this case we aimed to review systematically and thoroughly the literature of the use of PET imaging (amino acid PET imaging since there are plenty of evidence that have demonstrated their superiority compared to FDG) in the surgical management of glioma. The aim was not to support the use of PET imaging as an alternative to other techniques, but as contribution to give more meaningful contribution, since this imaging technique reflects the biological activity of the tumor.

I agree with the authors' idea of using amino acid PET to visualize the border of the invasive tumor, which is considered to be in between removal of the contrasted area and removal up to T2/FLAIR high signal and fuse it with surgical navigation. As the authors state, this removal procedure is not easy, and even if it becomes visible, whether or not the area can be removed depends on its localization.

It is debatable which is better, the supra-total approach of removing as much of the T2/FLAIR high signal area as possible and the boundaries of this area are determined by monitoring, or the authors' approach of removing as much of the amino acid PET high accumulation border as possible.

The result of the current review (namely the study by Inoue et al.), which was characterized by stringent criteria for inclusion, demonstrated that in most cases the PET area was smaller than T2/FLAIR hyperintensity. Furthermore, other references have showed that often PET active areas do not correspond to T2/FLAIR high signal areas. Therefore, the possible integration of PET imaging technique during resection, in combination with other techniques (i.e., MRI, intraoperative fluorophores, intraoperative neuromonitoring), does not pursue a minor amount of resected tissue compared to FLAIRectomy, but rather, to remove a “meaningful” area where hypothetically the glioma stem cells, which are linked to relapse, could be.

Therefore, even though amino acid PET clearly visualizes the extent of tumor involvement, there seems to be a lack of discussion on the advantages and disadvantages of applying amino acid PET to surgical excision. There is no disagreement that amino acid PET indicates the detection of a tumor, but it would be better to use amino acid PET as an area to increase the intensity of following radiotherapy (e.g., IMRT), or to communicate the removal rate to patients in order to collect more cases as the amino acid PET hyper-accumulation range.

The manuscript lacks these perspectives and seems unlikely to be supported by nuclear medicine specialists who have been pushing for PET scans. Even though many of the statements made in the discussion section are traces of existing imaging studies, such as what amino acid PET is able to visualize, they do not support the authors' conclusion that it finds utility in determining its role in surgery, particularly in determining the extent of resection. It would be better to rewrite it broadly and make it useful in determining the extent of treatment, by citing, for example, manuscripts that have been useful in determining the extent of radiotherapy.

Thank you for taking the time to review our manuscript and for your comment.

Unfortunately, we disagree with that. The current guidelines “Joint EANM/EANO/RANO practice guidelines/SNMMI procedure standards for imaging of gliomas using PET with radiolabeled amino acids and [18F]FDG: version 1.0”, supported by different figures involved in the management of patients affected by glioma (i.e. neuro-oncologists, neurosurgeons, radiotherapists, radiologists and nuclear medicine physicians), have agreed on the use of AA-PET imaging for the definition of the optimal biopsy site (i.e. site of maximum tracer uptake) and the delineation of tumor extent for surgery and radiotherapy planning.

In the discussion section there are only “traces” of studies in supporting our conclusion because only few studies have investigated the benefits conferred by integration of PET imaging (see table 2).

In these studies (2 out of 3 did not respect the selection criteria of our review for different reasons) the integration of PET and resection of PET active areas showed a significant improvement in terms of overall survival. Given these positive results, we investigated some reasons - by a neurosurgical point of view - that might have limited the diffusion of this imaging technique in surgical planning.

Finally, the review was intentionally focused on the use of AA-PET in neurosurgical practice, avoiding any digressing in other out-of-neurosurgeons’-concerns field.

minor comments.

Only PRISMA flow is described, which is in accordance with the guide for systematic reviews, but the content and the way it is summarized is hardly a systematic review. It is more acceptable to simply refer to it as a "literature review".

We respected the PRISMA criteria and as for other comments above, the systematic review concerned the use of PET imaging in the surgical practice, especially in term of resection (in fact, one of the search terms was “resection” for all field).

The references picked up in Tables 1 and 2 should be detailed, as there is no indication which part corresponds from PRISMA.

Given the stringent selection criteria and the lack of references that have met those criteria, we reported in table 1 some of those papers that passed to the second step of selection but not in the last step because they did not use PET imaging for selection but only for biopsy (as described in the table capture). While in table 2 we preferred to report not only the paper that met all criteria, but even other two papers (whose reasons for rejection are reported) where the integration of PET imaging showed a benefit in terms of OS.

Please give the references in the table a reference number.

We modified accordingly.

It is unclear which one manuscript corresponded to the last remaining manuscript in PRISMA.

We modified accordingly in the result section.

Round 2

Reviewer 1 Report (Previous Reviewer 2)

The authors have adequately responded to the raised comments with some English editing. The conclusion section is now reasonable and does not exceed the scope of this manuscript. Still, there are typos and errors, such as “PSF” instead of “PFS” in Line 40 and so on. If these minor errors had been fixed, I would agree to publish this manuscript.

Reviewer 3 Report (New Reviewer)

When treating malignant glioma (MG), it is very important to visualize and determine the true margin of the tumor.

In fact, MG always recurs, even if all the contrasted region of the MRI has been removed. Furthermore, when determining the target volume of radiotherapy, it is also required to define the extent of the tumor and the areas where the intensity of the treatment needs to be increased.

Surgical decision making using amino acid PET imaging has the potential to improve the prognosis of this disease.

The authors should review the PRISMA checklist one last time for the final manuscript.

This manuscript is a resubmission of an earlier submission. The following is a list of the peer review reports and author responses from that submission.

Round 1

Reviewer 1 Report

Degree of surgical resection for glioblastoma has been controversial. From an oncological point of view, the most a surgeon could resect safely is usually less than 1 log cell kill (if one considers both the contrast-enhancing and non-enhancing tumor). PET imaging in glioblastoma is likely best used to gauge treatment response and detect early recurrence.

Author Response

Reviewer #1: Degree of surgical resection for glioblastoma has been controversial. From an oncological point of view, the most a surgeon could resect safely is usually less than 1 log cell kill (if one considers both the contrast-enhancing and non-enhancing tumor). PET imaging in glioblastoma is likely best used to gauge treatment response and detect early recurrence.

Thank you for your comments and for taking the time to review our manuscript.

Degree of surgical resection for glioblastoma has been controversial in the past, but it is now universally accepted and demonstrated (as in the references cited in review, Ezquenazi et al (17), Certo et al (18), Eyüpoglu et al (102), Zigiotto et al (107)) how supratotal resection is associated with increased both overall survival and disease-free survival. The current challenge is indeed to customize the extent of resection, maximizing oncologic excision without neurologically harming the patient.

Certainly, at the present stage, the use of PET in glioblastoma is used to customize treatment in recurrences, assessing the actual presence of disease progression or early recurrence. However, the aim of our review is to evaluate its use in early diagnosis, with the goal of being more precise in defining tumor areas beyond contrast enhancing. This aspect is of fundamental importance for neurosurgical treatment in defining the onco-functional balance, with the aim of defining the borders of a supra total resection, which has already shown an effective benefit on survival, based not only on MRI FLAIR sequences.

Reviewer 2 Report

This study reviewed the role of amino-acids PET imaging specifically in improving the extent of resection of high-grade gliomas and accordingly increasing PFS and OS.  Although I agreed with their conclusions of the importance of amino acids PET imaging for the treatment of glioma, this review has several issues to be solved before publication

 The following topics should be discussed to demonstrate usefulness of amino-acids PET imaging in malignant glioma surgery.

#1How different is the ability of amino acids PET to detect tumors from conventional MRI?

#2How different is the ability to detect the tumor boundary among the tracers? 

#3How much has the surgical outcome changed with the navigation system that integrates PET and MRI?

#4What specific prospective studies should be done to prove the usefulness of amino acids PET in glioma surgery in the future? Or what kind of research is being done now?

Therefore, it is expected that a brief summary along these sub-themes will provide a good understanding of the current state of amino acids PET for the treatment of glioma.

Author Response

Reviewer #2: This study reviewed the role of amino-acids PET imaging specifically in improving the extent of resection of high-grade gliomas and accordingly increasing PFS and OS.  Although I agreed with their conclusions of the importance of amino acids PET imaging for the treatment of glioma, this review has several issues to be solved before publication

The following topics should be discussed to demonstrate usefulness of amino-acids PET imaging in malignant glioma surgery.

Thank you for taking the time to review our manuscript, we appreciate the opportunity to improve the quality of our work through your insightful comments. We have tried to answer point by point.

#1How different is the ability of amino acids PET to detect tumors from conventional MRI?

Lines 187 to 199 of the discussion section partially answer the question, particular the reference 24, in which it is shown how AA-PET can detect tumor boundaries with greater specificity and sensitivity. In any case, the following two paragraphs have been added detailing the differences between AA-PET and classical gadolinium MRI

In detail, AA-PET imaging has been demonstrated to identify infiltrating regions of tumor extending more accurately beyond the MRI contrast-enhancing lesion, delineating significantly larger tumor volumes, and to better define tumor boundaries within nonspecific regions of MRI T2/FLAIR signal abnormality (infiltrative disease vs vasogenic edema) [Yang Y, He MZ, Li T, Yang X. MRI combined with PET-CT of different tracers to improve the accuracy of glioma diagnosis: a systematic review and meta-analysis. Neurosurg Rev. 2019 Jun;42(2):185-195.]. In addition, AA-PET provides further insights regarding tumor heterogeneity, biologic activity, or aggressiveness of the disease [Zhang-Yin JT, Girard A, Bertaux M. What Does PET Imaging Bring to Neuro-Oncology in 2022? A Review. Cancers (Basel). 2022 Feb 10;14(4):879. doi: 10.3390/cancers14040879].

#2How different is the ability to detect the tumor boundary among the tracers? 

The following paragraphs and references have been added in response to this report

Among Met, FET and DOPA PET tracers, similar results in terms of accuracy have been reported regarding the ability to detect tumor boundary [*** Lapa C, Linsenmann T, Monoranu CM, Samnick S, Buck AK, Bluemel C, Czernin J, Kessler AF, Homola GA, Ernestus RI, Löhr M, Herrmann K. Comparison of the amino acid tracers 18F-FET and 18F-DOPA in high-grade glioma patients. J Nucl Med. 2014 Oct;55(10):1611-6. doi: 10.2967/jnumed.114.140608; **** Becherer A, Karanikas G, Szabó M, Zettinig G, Asenbaum S, Marosi C, Henk C, Wunderbaldinger P, Czech T, Wadsak W, Kletter K. Brain tumour imaging with PET: a comparison between [18F]fluorodopa and [11C]methionine. Eur J Nucl Med Mol Imaging. 2003 Nov;30(11):1561-7. doi: 10.1007/s00259-003-1259-1; ***** Dunet V, Rossier C, Buck A, Stupp R, Prior JO. Performance of 18F-fluoro-ethyl-tyrosine (18F-FET) PET for the differential diagnosis of primary brain tumor: a systematic review and Metaanalysis. J Nucl Med. 2012 Feb;53(2):207-14. doi: 10.2967/jnumed.111.096859.] When compared to Met and FET the main potential drawback of DOPA is its specific uptake in the striatum that may affect the ability to assess involvement of the putamen and caudate nucleus [*** Lapa C, Linsenmann T, Monoranu CM, Samnick S, Buck AK, Bluemel C, Czernin J, Kessler AF, Homola GA, Ernestus RI, Löhr M, Herrmann K. Comparison of the amino acid tracers 18F-FET and 18F-DOPA in high-grade glioma patients. J Nucl Med. 2014 Oct;55(10):1611-6. doi: 10.2967/jnumed.114.140608; ****** Kratochwil C, Combs SE, Leotta K, Afshar-Oromieh A, Rieken S, Debus J, Haberkorn U, Giesel FL. Intra-individual comparison of ¹⁸F-FET and ¹⁸F-DOPA in PET imaging of recurrent brain tumors. Neuro Oncol. 2014 Mar;16(3):434-40. doi: 10.1093/neuonc/not199.] On the other hand, DOPA uptake within the striatum gives the opportunity to further stratify tumoral uptake ratios through comparison with both the normal background levels and the striatum [****** Morana G, Puntoni M, Garrè ML, Massollo M, Lopci E, Naseri M, Severino M, Tortora D, Rossi A, Piccardo A. Ability of (18)F-DOPA PET/CT and fused (18)F-DOPA PET/MRI to assess striatal involvement in paediatric glioma. Eur J Nucl Med Mol Imaging. 2016 Aug;43(9):1664-72. doi: 10.1007/s00259-016-3333-5].

#3How much has the surgical outcome changed with the navigation system that integrates PET and MRI?

Since the majority of studies have investigated the pathologic “meaning” of the AA-PET positive tissue obtained with biopsy, only a few have analyzed the impact of performing a resection with integration of AA-PET imaging: although PET positive areas are often beyond the border of contrast enhanced nodule, no increase in neurological deficit has been reported.  in any case, the currently available scientific evidence is already not so clear-cut in proving or in refuting the usefulness of a PET image-guided resection integrated into an MRI-based neuronavigation system. Currently, it has come to be demonstrated and confirmed that PET positive areas beyond the enhancing nodule are indeed pathological, so the next step will be exactly to validate a PET aided resection.

Despite mentioned in the “only biopsy” group, the multicenter clinical trial by Wakabayashi et al [104] interestingly reported a modification of the extent of resection considering in the surgical planning the use of 18-flucovine-PET in 47.2% of cases in-creasing by 47.8% the resection of high-grade glioma.

Aiming the resection of AA-PET positive areas and BTV could be a good compromise between “FLAIRectomy” and classical GTR. However, as the functional result of the patient is the real surgeon’s aim, it is clear that an image-based resection has its well-known limitations (Duffau et al (105), Duffau et al (106), Zigiotto et al (107)]. 

#4What specific prospective studies should be done to prove the usefulness of amino acids PET in glioma surgery in the future? Or what kind of research is being done now?

As partially answered in the previous point, the next step will be to validate a PET aided resection. At our institution, we have initiated a scientific project for this very purpose, using 18-FDOPA PET preoperatively to complement gadolinium-enhanced MRI in intraoperative neuronavigation. The outcomes are both extent of resection and survivals, and histomolecular analysis of the excised parts.

Indeed, studies aimed at validating this strategy should target these outcomes, both anatomo-pathological to confirm and precisely define the microscopic extent of disease, and volumes resected and survivals to verify an actual clinical benefit of the technique.